# Molecular Imaging of Ultrasound-Mediated Blood-Brain Barrier Disruption in a Mouse Orthotopic Glioblastoma Model

**DOI:** 10.3390/pharmaceutics14102227

**Published:** 2022-10-19

**Authors:** Chiara Bastiancich, Samantha Fernandez, Florian Correard, Anthony Novell, Benoit Larrat, Benjamin Guillet, Marie-Anne Estève

**Affiliations:** 1CNRS, INP, Inst Neurophysiopathol, Aix-Marseille University, 13005 Marseille, France; 2Centre Européen de Recherche en Imagerie Médicale (CERIMED), CNRS, Aix-Marseille University, 13005 Marseille, France; 3APHM, CNRS, INP, Inst Neurophysiopathol, Hôpital Timone, Service Pharmacie, Aix-Marseille University, 13005 Marseille, France; 4CEA, CNRS, Inserm, BioMaps, Service Hospitalier Frédéric Joliot, Université Paris-Saclay, 91401 Orsay, France; 5CEA, CNRS, NeuroSpin/BAOBAB, Université Paris-Saclay, 91191 Gif-sur-Yvette, France; 6Centre de Recherche en Cardiovasculaire et Nutrition (C2VN), INSERM, INRAE, Aix-Marseille University, 13005 Marseille, France

**Keywords:** glioblastoma, focused ultrasound, blood-brain barrier, PET, SPECT, drug delivery

## Abstract

Glioblastoma (GBM) is an aggressive and malignant primary brain tumor. The blood-brain barrier (BBB) limits the therapeutic options available to tackle this incurable tumor. Transient disruption of the BBB by focused ultrasound (FUS) is a promising and safe approach to increase the brain and tumor concentration of drugs administered systemically. Non-invasive, sensitive, and reliable imaging approaches are required to better understand the impact of FUS on the BBB and brain microenvironment. In this study, nuclear imaging (SPECT/CT and PET/CT) was used to quantify neuroinflammation 48 h post-FUS and estimate the influence of FUS on BBB opening and tumor growth in vivo. BBB disruptions were performed on healthy and GBM-bearing mice (U-87 MG xenograft orthotopic model). The BBB recovery kinetics were followed and quantified by [99mTc]Tc-DTPA SPECT/CT imaging at 0.5 h, 3 h and 24 h post-FUS. The absence of neuroinflammation was confirmed by [18F]FDG PET/CT imaging 48 h post-FUS. The presence of the tumor and its growth were evaluated by [68Ga]Ga-RGD_2_ PET/CT imaging and post-mortem histological analysis, showing that tumor growth was not influenced by FUS. In conclusion, molecular imaging can be used to evaluate the time frame for systemic treatment combined with transient BBB opening and to test its efficacy over time.

## 1. Introduction

Glioblastoma (GBM) is the most aggressive and malignant primary brain tumor in adults. Today, GBM is still incurable and its effective treatment is hampered, among other factors, by the presence of the blood-brain barrier (BBB). The BBB limits the access of active agents at the tumor site and in proximity of the tumor, where it is intact but where infiltrating tumor cells reside and are responsible for recurrences [1]. The BBB is a physiological barrier protecting the central nervous system physically and biochemically from pathogens, toxins and hormones circulating in the blood. It is regulated by physical, transport, and metabolic properties of the endothelial cells surrounding the blood vessels, which are held together by tight junctions and reduce permeation of polar solutes from the blood to the brain extracellular compartment. The BBB permeability is also regulated by interactions with different vascular, immune, and neural cells [2]. Consequently, BBB is a highly selective filter through which nutrients needed by the brain are transmitted and waste products are removed. This protection of the brain complicates the treatment of many neurological diseases, such as GBM. Several strategies can be used to bypass the BBB to achieve enhanced delivery into the brain. Thus, for decades, intensive work has been done on methods to enable the transport of active substances into the brain by bypassing—or better still by selectively crossing—the BBB [3,4]. A range of strategies for overcoming the BBB have been developed for this purpose or are still under development [5,6,7]. Among these, the transient and local permeabilization induced by the application of focused ultrasound (FUS) to a selected region of the brain immediately following intravenous microbubbles administration is promising. The stable oscillation of microbubbles induced by FUS leads to biomechanical effects (cellular massage, microstreaming) on the endothelial cells composing the BBB wall causing a transient disruption of tight junction integrity and an increase of paracellular permeability. This leads to an increased extravasation of molecules that could not spontaneously cross the BBB (either free drugs or nanomedicines). The oscillating microbubbles can also induce an immune response by triggering the release of damage-associated molecular patterns [8]. This noninvasive technique has been shown to improve the brain delivery of therapeutic agents in several animal models [9], and to be safe and well-tolerated in clinical trials performed in patients with Alzheimer’s disease [10] and GBM [11,12]. Currently, at least two clinical trials are ongoing, testing single or multiple BBB disruptions during standard temozolomide chemotherapy in newly diagnosed GBM patients (clinicaltrials.gov ID NCT04614493 and NCT03712293 [13]). Others are ongoing on recurrent GBM patients in combination with other treatments (e.g., clinicaltrials.gov NCT03744026, NCT03626896, NCT04446416) showing a high interest of the neuro-oncology community for FUS-mediated BBB permeabilization.

Most of the preclinical studies reporting the use of this technique are performed under magnetic resonance (MR) guidance and use MR contrast agents (e.g., gadoterate meglumine, Gd-DOTA; Dotarem^®^) that do not cross the intact BBB to assess its disruption [14]. Indeed, molecules that do not have the physicochemical properties required to bypass the BBB are excellent markers of BBB integrity. In case of disease, brain lesion or induced-BBB opening, changes in the expression of BBB transporters and cell-cell junction proteins allow the increased delivery of contrast agents into the brain tissue making the BBB defects visible. Following this approach, Marty et al. used a quantitative T_1_ relaxometry method to follow BBB closure dynamics after FUS [15]. Unfortunately, the lack of sensitivity and rather low time resolution remains a limitation of MR based molecular imaging. By means of radioactively labelled tracers, which do not normally pass through the BBB, it is also possible to investigate the function of the BBB. This can be done using single photon emission computed tomography (SPECT) or positron emission tomography (PET). For example, in patients with acute stroke, increased uptake of 99mTc chelated with hexa-methyl-propylene-amine-oxime (HMPAO) [16,17] can be shown. A recent review by Arif et al. describes how PET can provide insights into the underlying mechanisms of FUS-mediated BBB permeabilization [18]. Despite its potential for this scope, the number of studies reporting the use of nuclear imaging techniques to evaluate the transient FUS-mediated BBB permeability, the enhanced intratumoral uptake of drug-loaded nanoparticles or the physiological effects induced by the transient BBB opening on the brain remains limited. A few studies have used nuclear imaging to establish optimal therapeutic windows for brain tumor chemotherapy following FUS-mediated BBB permeabilization and the systemic administration of theranostics nanoparticles or radiolabeled active agents in tumor-bearing or healthy animals ([111In]-labeled liposomal doxorubicin [19]; [89Zr]--cetuximab [20]).

In this study, we used nuclear imaging as an alternative to magnetic resonance imaging, which is not always available for preclinical studies. Our main objective was to use nuclear imaging to evaluate the effect of BBB opening on GBM-bearing mice. The secondary objectives were the evaluation the impact of BBB-opening on neuroinflammation and tumor growth. To do so, we evaluated the kinetics of transient FUS-mediated BBB opening via SPECT, together with computed tomography (CT) imaging using 99mTc-tagged diethylenetriamine pentaacetate acid ([99mTc]Tc-DTPA) [21] as a radiotracer. The neuroinflammation will be evaluated on GBM-bearing mice by PET/CT imaging using 18F-fluorodeoxyglucose ([18F]FDG) as a radiotracer [22,23] 48 h post-permeabilization. The tumor growth profile will be evaluated using 68Ga-labeled amino acid sequence arginine-glycine-aspartic ([68Ga]Ga RGD_2_) via PET/CT [24]. A survival analysis will be performed to evaluate if BBB permeability in the early stages of tumor development (before day 10 post-grafting [25]) can induce a delay in tumor growth. To the best of our knowledge, we are the first to use these radiotracers sequentially and on the same GBM-bearing animal.

## 2. Materials and Methods

The in vivo experiments reported in this work have been approved by the institution’s Animal Care and Use Committee (CE71, Aix-Marseille Université, reference n° 14713) and performed following the French national regulation guidelines in accordance with EU Directive 2010/63/EU. Mice were housed in enriched cages placed in a temperature- and hygrometry-controlled room, had free access to water and food and were monitored daily. The mice were six-week-old female athymic Nude-Foxn1nu mice (Envigo, Gannat, France). The animals were sacrificed at the end of the study (healthy animals) or when they reached the end-points (≥20% body weight loss or 10% body weight loss plus clinical signs of distress e.g., paralysis, arched back, or lack of movement for glioma-bearing animals).

### 2.1. Pilot Study on Healthy Animals

Ultrasound waves were produced using a spherically focused single element ultrasound transducer (central frequency 1.5 MHz, diameter 25 mm, focal depth 20 ± 2 mm, Imasonic, Voray sur I’Ognon, France) driven by a built-in signal generator connected via a 50 W power amplifier (Image Guided Therapy, Pessac, France). The output pressure of the transducer was measured in a degassed water tank, using a 0.2 mm lipstick hydrophone (HGL, ONDA Corporation, CA, USA) mounted on a positioning stage. The acquired signal from the calibrated hydrophone was sampled by the scope (Picoscope 5243A, Pico Technology, Eaton Socon, UK). Electrical power sent to the transducer was monitored in real-time during the sonication. The focal point size (e.g., 6 dB pressure focal region) of the transducer was 1.2 × 1.2 × 6 mm^3^.

Animals underwent the permeabilization protocol either for a single spot opening (1.2 × 1.2 mm) or a large brain opening (6 × 6 mm raster scan trajectory): in either case, they were anesthetized, placed on a temperature-monitored stereotactic device, and injected with microbubbles before launching the appropriate sequence.

Mice were anesthetized by ketamine/xylazine (100 and 10 mg/kg intraperitoneal injection, respectively) and placed in a dedicated temperature-controlled frame in a prone position under the set-up. The ultrasound transducer was coupled to the head via a water-filled latex balloon (Comed; Strasbourg, France) and degassed echographic gel (70% *v*/*v* in water). A hundred μL of SonoVue^®^ microbubbles (Bracco, Milan, Italy) were systemically administered via retro-orbital injection in the right eye using an insulin syringe (27G), and the ultrasound sequence was started immediately after. Depending on the experimental group, the transducer was still during the sonications or mechanically moved following programmed trajectories using a 2D-axis motorized positioning stage. Group A: single spot sonication (pulse length of 3 ms every 100 ms for 2 min) (*n* = 3); Group B: a 6 × 6 mm raster scan mechanical trajectory was designed to cover most of the brain (*n* = 3). As described in [20,26], ultrasonic waves were quasi-continuously transmitted at 1.5 MHz during transducer motion except during the changes of direction in the raster scan (duty cycle 69%). This avoided an excessive deposit of ultrasound energy at these locations. The sequence was repeated 25 times for a total exposure of 127 s. The transmitted in situ peak negative pressure in the mouse brain is estimated to be 430 kPa (520 kPa in deionized water) considering a skull attenuation of 18% (average value of the skull attenuation measured in [27]) at 1.5 MHz. One sham animal, undergoing the same procedure as the animals of group A and B (anesthesia, stereotactic fixing, microbubbles injections, SPECT imaging protocol) but without launching the mechanical scan and BBB permeabilization sequence was also included in the pilot study and imaged.

### 2.2. Hemispheric Blood-Brain Barrier Permeabilization on Healthy and Tumor-Bearing Mice

#### 2.2.1. Glioma Cell Cultures

U-87 MG glioma cells (ATTC, Manassas, FL, USA) were cultured in Eagle’s Minimum Essential Medium (EMEM; Gibco, Life Technologies, Carlsbad, CA, USA). Culture medias were supplemented with 10% Fetal Bovine Serum (FBS; Gibco, Life Technologies), 100 U/mL penicillin G sodium and 100 μg/mL streptomycin sulfate (Gibco, Life Technologies). Cells were subcultured in 75 cm^2^ culture flasks (Corning^®^ T-75, Sigma-Aldrich, Saint-Louis, MO, USA) and incubated at 37 °C and 5% CO_2_.

#### 2.2.2. Orthotopic U-87 MG Human Glioblastoma Tumor Model

Animals were anesthetized by ketamine/xylazine (100 and 10 mg/kg intraperitoneal injection, respectively) and fixed in a stereotactic frame. The skin surface on the head was disinfected by application of an antiseptic solution (Vétédine^®^ solution, Vetoquinol, Lure, France). Lidocaine (10 mg/mL; Aguettant, Lyon, France) was injected subcutaneously at the site of incision, and the eyes were protected with an ophthalmic gel (Ocry-gel, TVM lab, Lempdes, France). To hydrate the animal, 200 μL of physiological solution (0.9% Sodium Chloride; Aguettant, France) were injected subcutaneously in the flank. An incision was made along the midline and a burr hole was drilled into the skull at the right frontal lobe, 0.5 mm anterior and 2 mm lateral to the bregma (high speed drill: Tack life Tools, New York, NY, USA; 0.8 mm diameter round end engraving burrs: Dremel, Breda, The Netherlands). A 10 μL 26 s gauge syringe with cemented 51 mm needle (Hamilton, Rungis, France) was used to inject 5 × 10^4^ U-87 MG glioma cells suspended in EMEM (without FBS and antibiotics) at a depth of 3 mm from the outer border of the brain, using an automatic pump device at a speed of 0.7 μL/min. The wound was then closed using a tissue adhesive glue (3M Vetbond^®^, Sergy-Pontoise, France) and the animals recovered under an infrared heating lamp.

#### 2.2.3. Blood-Brain Barrier Permeabilization Protocol

Eight days following tumor grafting, mice were anesthetized by the intraperitoneal injection of ketamine/xylazine (100 and 10 mg/kg, respectively) and placed in a dedicated stereotactic frame positioned below the BBB opening set up as previously explained. For this experiment the mice were divided in three groups (*n* = 4–5 per group): Group “GBM + FUS”: tumor-bearing animals undergoing FUS-mediated hemispheric BBB permeabilization (3.6 × 3.6 mm trajectory covering the whole right hemisphere; quasi-continuous sonication—duty cycle 71%—repeated 36 times and paused between each execution and a moving speed of 10 mm/s; the total sonication time of 115 s [28]); Group “GBM only”: tumor-bearing animals undergoing the same procedure as group “GBM + FUS“ (anesthesia, surgical procedure, microbubbles injections, SPECT/PET imaging protocols) but without launching the mechanical scan and BBB permeabilization sequence (tumor growth control); Group “FUS only”: healthy animals undergoing FUS-mediated hemispheric BBB permeabilization (BBB opening control). Groups “GBM + FUS” and “GBM only” were imaged for [99mTc]Tc-DTPA, [18F]FDG and [68Ga]Ga-RGD_2_. The animals of group “FUS only” received all the surgical procedures of the group “GBM + FUS” the day of the permeabilization and were imaged for [99mTc]Tc-DTPA and [18F]FDG. Figure 1 recapitulates the experimental plan used for the FUS-mediated hemispheric BBB permeabilization study on healthy and tumor-bearing animals.

Mice were sacrificed when reaching the clinical end points (groups “GBM + FUS” and “GBM only”) or after the last imaging session (group “FUS only”). For tumor-grafted animals, statistical analysis was estimated from comparison of Kaplan-Meier survival curves using the log-rank test (Mantel Cox test). The brains of these animals were extracted and fixed in 10% formalin solution (Merck, Darmstadt, Germany) for 24 h before being rinsed in PBS and kept at 4 °C until further use. Brains were then embedded in paraffin, sectioned at 4 μm thickness using a MICROM HM 335 E microtome (Thermo Fischer Scientific, Waltham, MA, USA), collected on Silane adhesive KF Frost slides (VWR, Amsterdam, The Netherlands) on a drop of glycerate albumin (DiaPath, Martinengo, Italy) mixed with distilled water and dried on a Leica HI1220 heating plate (Leica, Wetzlar, Germany). Slides were incubated at 37 °C overnight and then stored at room temperature until further use. For standard histology, the slides were deparaffinized and stained with hematoxylin and eosin (H&E) for tumor detection (*n* = 3).

### 2.3. Imaging

#### 2.3.1. Single Photon Emission Computed Tomography (SPECT) Imaging

For mice included in the studies described in Section 2.1 and Section 2.2, the BBB opening was evaluated by SPECT/CT imaging using [99mTc]Tc-DTPA as radiotracer 0.5 h, 3 h and 24 h post BBB permeabilization. For animals of the study described in Section 2.2, this procedure was also performed 24 h before BBB permeabilization.

[99mTc]Tc-DTPA was administered via an intravenous injection of 50 μL of radioactive tracer (20 MBq) in isotonic and pyrogen-free solution using an insulin syringe (27G). Thirty minutes after injection, the SPECT/CT acquisition was done for 20 min under anesthesia with 1.5% vol% isoflurane (IsoVet^®^, Laboratoire Osalia, Paris, France) using a NanoSPECT/CTplus^®^ camera and the Nucline^®^ 1.02 acquisition software (Mediso Medical Imaging System Ltd., Budapest, Hungary). SPECT and CT DICOM files were fused for reconstruction and image processing was carried out with VivoQuant^®^ 3.5 and InvivoScope^®^ 2.00 reconstruction software (InviCRO, Boston, MA, USA) to assess tracer uptake in the brain.

For the pilot study, statistical analysis was performed using two-way ANOVA (uncorrected Fisher’s LSD test). For the hemispheric BBB permeabilization study, statistical analysis was performed using two-way ANOVA with Dunnett’s multiple comparisons test of each time point vs. the basal value (−24 h).

#### 2.3.2. Positron Emission Tomography (PET) Imaging

For the study described in Section 2.2, PET/CT imaging was also performed.

The presence, location and growth of the tumors were determined by PET/CT imaging using [68Ga]Ga-RGD_2_ as a radiotracer. Imaging was performed for all tumor-bearing mice included in the study at 7-, 17- and 24-days post tumor cell implantations (day −1, 9 and 16 before/after BBB permeabilization). Neuroinflammation was estimated by PET/CT imaging 48 h following sonication using [18F]FDG as a radiotracer. Statistical analysis was performed using one-way ANOVA test.

[68Ga]Ga-RGD_2_ or [18F]FDG were administered via an intravenous injection of 50 μL of radioactive tracer (5 MBq) in isotonic and pyrogen-free solution using an insulin syringe (27G). Forty-five minutes ([18F]FDG) or one hour ([68Ga]Ga-RGD_2_) after injection, the PET/CT acquisition was done for 20 min under anesthesia with 1.5% vol% isoflurane using a NanoScanPET/CT^®^ camera and the Nucline^®^ 1.02 acquisition software (Mediso Medical Imaging System Ltd., Budapest, Hungary). PET and CT DICOM files were fused for reconstruction and image processing was carried out with VivoQuant^®^ 3.5 and InvivoScope^®^ 2.00 reconstruction software (InviCRO, Boston, MA, USA). A statistical analysis was performed using a two-way ANOVA (uncorrected Fisher’s LSD test).

Quantitative analysis of the region of interest (ROI) of the PET or SPECT signal was performed on attenuation- and decay-corrected PET/SPECT images using InterviewFusion software (Mediso, Budapest, Hungary). Tissue uptake values for each mouse were expressed as an average ratio of the signal from each ipsilateral hemisphere to the contralateral signal ± SEM and in total brain as an average percentage of the injected dose per cubic millimetre of tissue (%ID/mm^3^) ± SEM.

## 3. Results and Discussion

### 3.1. Pilot Study on Healthy Animals

To understand the optimal permeabilization parameters and time frames for SPECT/CT imaging with [99mTc]Tc-DTPA, a FUS-mediated BBB-opening pilot study was performed on healthy animals. Animals underwent the permeabilization protocol either for a single spot opening (1.2 × 1.2 mm) or a large brain opening (6 × 6 mm raster scan trajectory); the sequences were selected based on previous results within our team of collaborators [15,29,30,31] and adapted to the characteristics of our transducer and animal species and strain (nude mice). For these experiments, a multi-functional preclinical device recently assembled for combined BBB permeabilization and photothermal therapy with photoacoustic temperature monitoring was used [32].

Immediately following sonication, animals were administered with [99mTc]Tc-DTPA and underwent SPECT/CT imaging 0.5 h later under isoflurane mild anesthesia (Figure 2). SPECT/CT imaging was also performed 3 h and 24 h later to evaluate the BBB closure kinetic.

For both sonication schemes (single spot and large brain opening), animals did not show any sign of pain or distress following BBB permeabilization. In both groups, an increase in the [99mTc]Tc-DTPA SPECT/CT signal was observed in the brain in the treated area and showed a peak 0.5 h following BBB disruption (0.5 h vs. 24 h: 3.9-fold increase in single spot scheme, *p* = 0.0289; 7-fold increase in the raster scan scheme, *p* = 0.0003), followed by a decrease of the signal at 3 h (3 h vs. 24 h: 3.1-fold increase in single spot scheme, *p* = 0.0979; 5.6-fold increase in the raster scan scheme, *p* = 0.0016) and returning to basal levels comparable to the sham animal at 24 h after FUS exposure. As expected, a higher increase (two-fold time at 0.5 h and 3 h) in the total amount of [99mTc]Tc-DTPA was observed in the raster scan scheme in comparison with the animals that underwent single spot sonication (single spot vs. raster scan trajectory: *p* = 0.0144 at 0.5 h; *p* = 0.0379 at 3 h; *p* = 0.9290 at 24 h) as the exposed volume in the latter is much lower.

These results are in accordance with previous works showing that BBB closure following FUS permeabilization is gradual and depends on the size of the contrast agent injected [15]. As Dotarem^®^ has a spherical hydrodynamic diameter similar to the one of [99mTc]Tc-DTPA (approximately 1 nm), a signal up to 24 h following sonication was expected as suggested in [15]. However, at 3 h, the amount of quantified [99mTc]Tc-DTPA was already decreasing, going back to basal values at 24 h. Variations in the experimental protocol (e.g., animal model, microbubbles concentration, FUS protocol and acoustic pressure) as well as physicochemical differences between the contrast agents could also explain the differences observed between previously reported studies. For example, one of the factors that could explain the different brain uptake between Dotarem^®^ and [99mTc]Tc-DTPA is the different anesthetics used for our experiment compared to Marty et al. (ketamine/xylazine plus isoflurane in medical air only vs. isoflurane in a mixture of air and oxygen) [15]. Indeed, it has been previously demonstrated that the circulation time of the microbubbles in the presence of oxygen (used as carrier gas for inhalation anesthesia) and the different vasoactive effect of anesthetic protocols can modify BBB opening efficacy, limiting the comparison between experimental data obtained in different labs [33,34].

If the objective of BBB opening for GBM treatment is to increase the amount of drug reaching the tumor, the single spot scheme would be the most appropriate of the two, as it allows a localized BBB permeabilization around the tumor site, potentially increasing its efficacy while avoiding local side effects in other parts of the brain [15]. However, in our setup the animal is placed on the stereotactic frame but the control of the exact position of the transducer remains challenging. Indeed, contrarily to the MRI-guided BBB disruption protocol, where acoustic radiation force imaging (ARFI) can be performed before the injection of microbubbles to locate the ultrasound focal spot [35], our device does not allow such targeting accuracy. Only at the end of the experiment could signals obtained by [99mTc]Tc-DTPA-mediated SPECT be compared with the ones obtained by [68Ga]Ga-RGD_2_-mediated PET/CT to evaluate if the opening was successfully performed around the tumor site. On the other side, the amount of drug (or tracer, in this case) that can reach the brain and then diffuse to the tumor site is much higher for the large brain sonication scheme. As in the configuration we used there is not enough brain left “untreated”, quantification errors can be performed as the measures do not consider the interindividual heterogeneity. Indeed, each mouse presents a different basal value following the administration of contrast agents. Because of this inter-individual variability between different mice, the quantifications of radiolabeled tracer uptake for the whole brain experiments are challenging and can lead to errors, as no baseline can be acquired. To overcome these limitations and increase the amount of [99mTc]Tc-DTPA reaching the tumor, we decided to perform the next series of experiments using a one hemisphere only BBB permeabilization scheme. This would allow us to make sure to open the BBB around the tumor while using as an internal control the contralateral “untreated” hemisphere for the quantification of the imaging tracers.

### 3.2. Hemispheric Blood-Brain Barrier Permeabilization and Imaging Study on Healthy and Tumor-Bearing Animals

Our purpose is to show the interest in using nuclear imaging to evaluate the efficacy and effect of an FUS-mediated BBB opening in healthy and tumor-bearing animals. To do so, we performed a study on animals with or without tumor grafting and exposed or not to hemispheric BBB permeabilization (see experimental plan in Figure 1). As for the pilot study, animals did not show any sign of pain or distress following the hemispheric permeabilization protocols. As shown in Figure 3, [99mTc]Tc-DTPA quantification confirmed the transient BBB opening on the hemisphere exposed to FUS on tumor-bearing animals (group “GBM + FUS”) and healthy animals (group “FUS only”), with an increase in ipsilateral/contralateral ratio at times 0.5 and/or 3 h and a return to basal signal within 24 h (group “GBM + FUS”: −24 h vs. 0.5 h *p* = 0.0399; −24 h vs. 3 h *p* = 0.0301; −24 h vs. 24 h *p* = 0.1727; group “FUS only”: −24 h vs. 0.5 h *p* = 0.0247; −24 h vs. 3 h *p* = 0.4162; −24 h vs. 24 h *p* = 0.5683; raw quantification data Appendix A). These results also suggest that the tumor may have an impact on BBB permeabilization, which appears to be greater and longer in tumor-bearing animals than in healthy animals. The results of the “GBM only” group show that the BBB is not permeable to the tested tracer ([99mTc]Tc-DTPA) in GBM-bearing animals at the time of FUS-mediated opening (eight days post U-87 MG grafting). However, modifications in the composition and activation state of microglia and astrocytes and their interaction with pericytes and endothelial cells around the tumor lesion might lead to a prolonged effect of FUS-mediated BBB opening in the “GBM + FUS” group. Moreover, the increased mechanical stiffness of the tumor region as well as the cross-talk between healthy cells, tumor cells and other components of the tumor microenvironment (e.g., tumor associated macrophages) might also alter the extravasation of contrast agent after sonopermeabilization leading to a slower recovery following FUS-mediated BBB opening. Therefore, our results suggest that the consequences of tumor growth on its reaction to FUS-induced BBB stress will have to be investigated further.

Imaging techniques able to show the initiation and propagation of neuroinflammation (ex. BBB permeability, leukocyte infiltration, microglial activation, and upregulation of cell adhesion molecules) in real-time on living animals are useful to evaluate the consequences of BBB disruption on tumor growth and treatment outcomes. Immunocompetent CNS cells or peripheral immune cells can be used as neuroinflammatory targets [36]. For example, iron oxide nanoparticles that can be internalized by circulating monocytes or by macrophages following extravasation can be used as MRI contrast agents. This cell labelling technique should make it possible to monitor the evolution of the inflammatory reaction in the brain in real time in a non-invasive manner. The conjugation of the nanoparticles with ligands (ex. peptides, proteins, or antibodies) can increase the specificity of cellular uptake to target cells expressing cellular markers of neuroinflammation [37,38]. In this work, nuclear imaging of neuroinflammation was done with [18F]FDG, as this PET radiotracer is currently used in the clinics. The availability of compounds other than commercial radiopharmaceuticals is very limited in most PET centers. Other PET tracers are in preclinical development in our and other groups, but their use is costly and their clinical application still uncertain.

[18F]FDG is commonly used for oncologic imaging as tumor cells possess increased energy demand and elevated metabolism, but it also shows high uptake in inflammatory cells [39]. Brendel et al. correlated aging hypermetabolism and neuroinflammation using FDG and 18F-GE180 (a tracer for the 18-kDa translocator protein TSPO, which is highly expressed in activated microglia) PET imaging in conjunction with biochemical assessments of neuroinflammatory markers [40], thus validating this radiotracer for this use also in patients [41]. The physiologic uptake of FDG by cells is related to the rate of glucose metabolism and glucose transporters expression, varying in normal structures, inflammatory sites, and tumors. Several studies have shown that by modifying FDG-PET analysis methods or acquisition timings it is possible to differentiate inflammation from tumor malignancies. For example, Yang et al. have used dynamic FDG-PET in a mouse model of non-small cell lung carcinoma [42] for this purpose, while Hustinx et al. used dual time point FDG-PET in head and neck cancers as FDG uptake over time shows a different pattern in benign and malignant tissues [43]. Verhoeven et al. have used conventional (60 min post-injection) and delayed (240 min post-injection) FDG-PET in U-87 MG GBM tumors and turpentine-invoked flank inflammation to evaluate the ability of this tracer to differentiate tumor from necrosis. Their results showed that only delayed [18F]FDG PET was also able to discriminate GBM from radiation necrosis, while both conventional and delayed [18F]FDG display significant uptake in the turpentine-invoked lesion [44]. In our study, to assess inflammatory uptake and not the malignant lesion, we performed [18F]FDG PET/CT 45-min following tracer injection. Our results show that the BBB opening did not induce differences in brain metabolism in any of the groups as demonstrated by the [18F]FDG PET/CT imaging results 48 h post-permeabilization, suggesting the absence of neuroinflammation in this experimental protocol (Figure 4 and Appendix A).

Lastly, [68Ga]Ga-RGD_2_ PET/CT imaging showed an increase over time for tumor-bearing animals, representing tumor growth, without significant differences between mice that were exposed to FUS-mediated BBB opening and control mice (day -1 *p* = 0.3554; day 9: *p* = 0.2871; day 16 *p* = 0.8780; Figure 5). Indeed, no direct RGD_2_ brain uptake due to FUS was expected since RGD_2_ imaging was performed late after BBB recovery. Thus, this confirms that a single session BBB opening does not seem to affect (negatively nor positively) α_ν_β_3_ integrin (the well-known target of RGD peptide) expression in our tumor model. No difference in the animals’ survival was observed between groups “GBM + FUS“ and “GBM only” (as shown in Figure 6A). The H&E staining shown in Figure 6B shows the presence of relatively low-enhancing glioblastoma tumor cells forming spheric tumor lesions neatly divided from the surrounding normal brain tissue both in the “GBM only” and “GBM + FUS” groups. In both cases, the tumor grows in one hemisphere suggesting that BBB-opening in early stages of tumor development does not induce a delay in tumor growth nor tumor cells infiltration in healthy regions of the brain.

Lin et al. were the first to investigate the feasibility of micro-SPECT/CT and [99mTc]Tc-DTPA for identifying the disruption of the BBB induced by FUS in healthy Sprague-Dawley rats [21]. The sonication was performed after craniotomy (removing skull), directly on the brain to reduce the distortion of the ultrasonic focal beam. Static SPECT imaging showed a peak in [99mTc]Tc-DTPA signal 1.5 h post-sonication and injection of the tracer, and they compared the extent and intensity of radioactivity with autoradiography and histology. The authors reported that high acoustic powers allowed the delivery of higher amounts of radiotracer, but brain hemorrhage occurred at pression amplitudes higher than 1.9 MPa. Yang et al. evaluated the pharmacokinetics of [99mTc]Tc-DTPA after systemic administration in healthy or F98 glioma-bearing F344 rats with or without FUS-mediated BBB permeabilization of one hemisphere of the brain [45]. In both cases, animals were anesthetized by isoflurane (2% in 100% oxygen) and the tracer was co-injected with Evans Blue to quantify the BBB permeability by its extravasation rate. The maximum peak of radioactivity was observed within one hour from BBB permeabilization. To evaluate tissue damage, the authors performed histological analysis after sacrifice of the animals (about 4 h post-sonication) showing no significant differences in apoptosis between sonicated tumors and control tumors. This study demonstrated that [99mTc]Tc-DTPA microSPECT/CT can be used for pharmacokinetic analysis of FUS-induced BBB disruption. However, they did not evaluate the long-term consequences of the permeabilization on tumor growth, which is what we did in our study. The same authors evaluated the pharmacokinetics of [18F]FDG by dynamic PET and expression of glucose transporter 1 (GLUT1) protein by western blot analysis after FUS-mediated BBB disruption in healthy animals (0 h and 21 h post-sonication) [46]. Their results showed a decrease in glucose uptake and GLUT1 protein expression following transient BBB opening, which recovers after 24 h. In our study, we confirm the absence of difference in [18F]FDG uptake 48 h post-permeabilization. Okada et al. used 2-amino-[3-^11^C]isobutyric acid ([3-^11^C]AIB) as a PET probe to quantify BBB disruption (lipopolysaccharide-mediated or FUS-mediated) in healthy rats [47]. They compared their PET results with autoradiography and Evans Blue coloration ex vivo (for lipopolysaccharide-mediated BBB opening) and Gd-DTPA-enhanced MRI in vivo (for FUS-mediated BBB opening), showing the usefulness of this radiotracer for noninvasive BBB permeability measurement. The difference of kinetics observed between Gd-DTPA and [3-^11^C]AIB in the brain can be explained by the difference in transport into the brain cells rather than a difference in the blood kinetics between the two agents. In addition, our collaborators recently reported the use of [18 F]-2-fluoro-2-deoxy-sorbitol ([18 F]-FDS) PET imaging to evaluate BBB integrity following FUS-induced permeabilization in healthy animals [28]. Brain distribution of [18 F]-FDS was consistent with ex vivo Evans Blue extravasation validating this tracer as a quantitative marker of BBB permeability. As previous studies have shown the accumulation of [18 F]-FDS in orthotopic GBM tumors in mice [48], we did not choose this radiotracer to evaluate the kinetic of BBB permeabilization in our work. Sultan et al. evaluated the effect of surface charges of ^64^Cu-integrated ultrasmall gold nanoclusters on brain distribution following FUS-mediated BBB permeabilization by PET/CT imaging [49]. They showed that neutrally charged nanocarriers perform the best in terms of theranostic delivery to the brain. Finally, Sinharay et al. used PET/CT imaging with [18F]-DPA714 (a biomarker of translocator protein), to assess neuroinflammatory changes in healthy animals 24 h or 1–2 weeks following single or multiple FUS-mediated BBB-permeabilization, respectively [50]. In this study performed on healthy rats, nuclear imaging was not used to evaluate BBB opening as the authors used MRI for this purpose. The neuroinflammation reported by PET/CT imaging was confirmed by histology (microglial activation by Iba1 staining; astrocytosis by GFAP staining) and seemed to persist for up to two weeks, with no evidence of cumulative inflammatory effect following multiple BBB-opening. In our study, we did not observe neuroinflammation 48 h after single BBB-opening with the [18F]FDG tracer. Immunophenotyping, morphological characterization, gene expression and phosphorylation levels of inflammatory proteins could be used in the future to confirm our results ex vivo (e.g., via flow cytometry, immunocytochemistry, quantitative RT-PCR and proteomics) [51].

## 4. Conclusions

In conclusion, molecular imaging using SPECT/CT and PET/CT allowed us to evaluate FUS-mediated BBB disruption and closure in mice bearing GBM tumors. By using three radiotracers ([99mTc]Tc-DTPA, [18F]FDG and [68Ga]Ga-RGD_2_)sequentially and on the same GBM-bearing animal, we also demonstrated that our experimental protocol is safe (lack of neuroinflammation) and that tumor growth is not influenced by the BBB opening. This study could be relevant for future clinical applications as [99mTc]Tc-DTPA and [18F]FDG are currently used in patients and FUS-mediated BBB opening is tested in GBM clinical trials.

Our results show that rodent PET/SPECT imaging can be an efficient tool to evaluate the time frame for systemic treatment combined with transient BBB opening, and to test its efficacy over time. This versatile imaging modality allows us to follow the pathological processes and the opening efficiency with high sensitivity. Further characterization using more infiltrating GBM rodent models and assessing the impact of FUS-mediated BBB opening on the brain and tumor microenvironment could be of interest. Moreover, the use of nuclear imaging in combination with more commonly used imaging techniques such MRI could allow the optimization of drug delivery protocols.

## Figures and Tables

**Figure 1 pharmaceutics-14-02227-f001:**
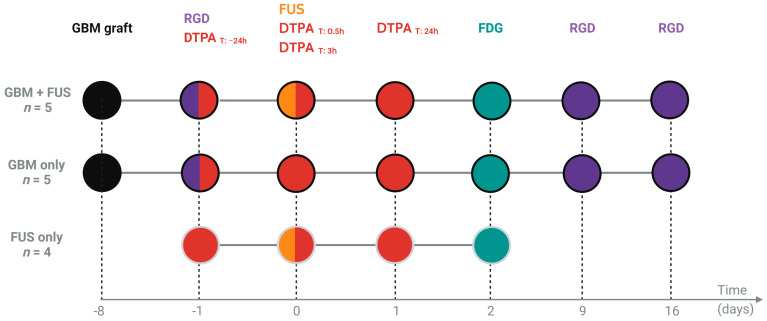
Schematic representation of the experimental plan used for the FUS-induced hemispheric BBB permeabilization and imaging study on healthy and tumor-bearing animals. Group “GBM + FUS“: tumor-bearing animals undergoing FUS-mediated BBB permeabilization; Group “GBM only”: tumor-bearing animals without BBB permeabilization (tumor growth control); Group “FUS only”: healthy animals undergoing FUS-mediated BBB permeabilization (BBB opening control).

**Figure 2 pharmaceutics-14-02227-f002:**
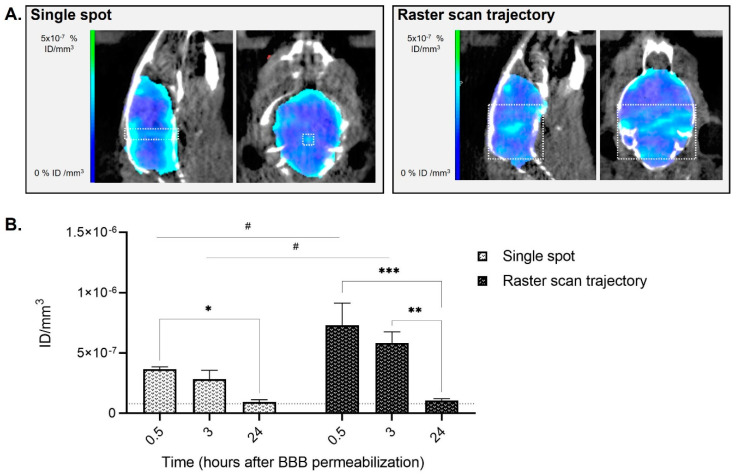
Imaging of the animal’s brain following BBB disruption and [99mTc]Tc-DTPA intravenous injection in healthy mice 0.5 h later. (**A**) Brain representative SPECT/CT tomographic images of [99mTc]Tc-DTPA distribution in animals that received single-spot (group A, left panel) or raster scan trajectory (group B, right panel) focused ultrasound. The white square represents the approximate region were FUS were applied for both sonication schemes; (**B**) Quantification of [99mTc]Tc-DTPA activity 0.5 h, 3 h and 24 h following focused ultrasound. Results are expressed as injected dose per tissue volume (ID/mm^3^; *n* = 3; mean ± SEM). Statistical differences are reported as # for comparisons between single spot vs. raster scan trajectory and as * for comparisons between 0.5 h vs. 3 h vs. 24 h (^#^
*p* < 0.05; * *p* < 0.05; ** *p* < 0.01; *** *p* < 0.001). The black dotted line is a visual representation of the baseline of an animal that did not receive BBB permeabilization, obtained as an average of the 0.5 h, 3 h and 24 h [99mTc]Tc-DTPA quantifications of the sham animal (7.8 × 10^−8^ ± 1.6 × 10^−8^).

**Figure 3 pharmaceutics-14-02227-f003:**
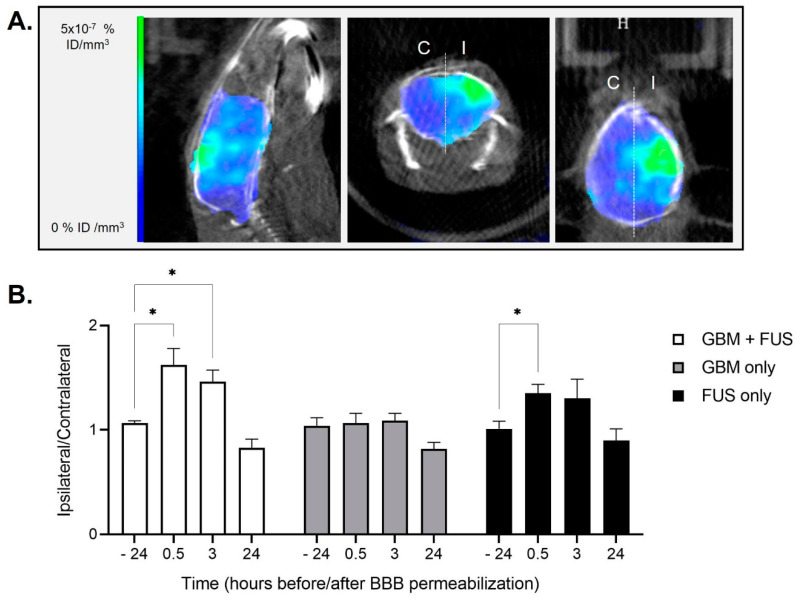
BBB permeabilization and [99mTc]Tc-DTPA SPECT/CT imaging on healthy and tumor-bearing animals. (**A**) Brain representative images obtained by [99mTc]Tc-DTPA-mediated SPECT/CT imaging in an animal of group “GBM + FUS” (tumor-bearing animals undergoing FUS-mediated BBB permeabilization) 0.5 h after hemispheric trajectory focused ultrasound. The white dotted line represents the separation between ipsilateral (I) and contralateral (C) brain hemispheres that was used for the quantifications; (**B**) Quantification of [99mTc]Tc-DTPA activity in animals of group “GBM + FUS” (tumor-bearing animals undergoing FUS-mediated BBB permeabilization, *n* = 5), group “GBM only” (tumor-bearing animals without FUS-mediated BBB permeabilization, *n* = 5) and group “FUS only” (healthy animals undergoing FUS-mediated BBB permeabilization, *n* = 4). Results are expressed as ipsilateral/contralateral ratio (mean ± SEM; * *p* < 0.05).

**Figure 4 pharmaceutics-14-02227-f004:**
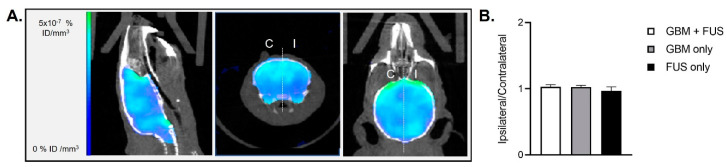
BBB permeabilization and [18F]FDG PET/CT imaging on healthy and tumor-bearing animals. (**A**) Brain representative images of [18F]FDG-mediated PET/CT two days after hemispheric trajectory focused ultrasound in an animal of group “GBM + FUS” (tumor-bearing animals undergoing FUS-mediated BBB permeabilization). The white dotted line represents the separation between ipsilateral (I) and contralateral (C) brain hemispheres that was used for the quantifications; (**B**) Quantification of [18F]FDG activity in animals of group “GBM + FUS” (tumor-bearing animals undergoing FUS-mediated BBB permeabilization, *n* = 5), group “GBM only” (tumor-bearing animals without FUS-mediated BBB permeabilization, *n* = 5) and group “FUS only“ (healthy animals undergoing hemispheric FUS-mediated BBB permeabilization, *n* = 4). Results are expressed as ipsilateral/contralateral ratio (mean ± SEM).

**Figure 5 pharmaceutics-14-02227-f005:**
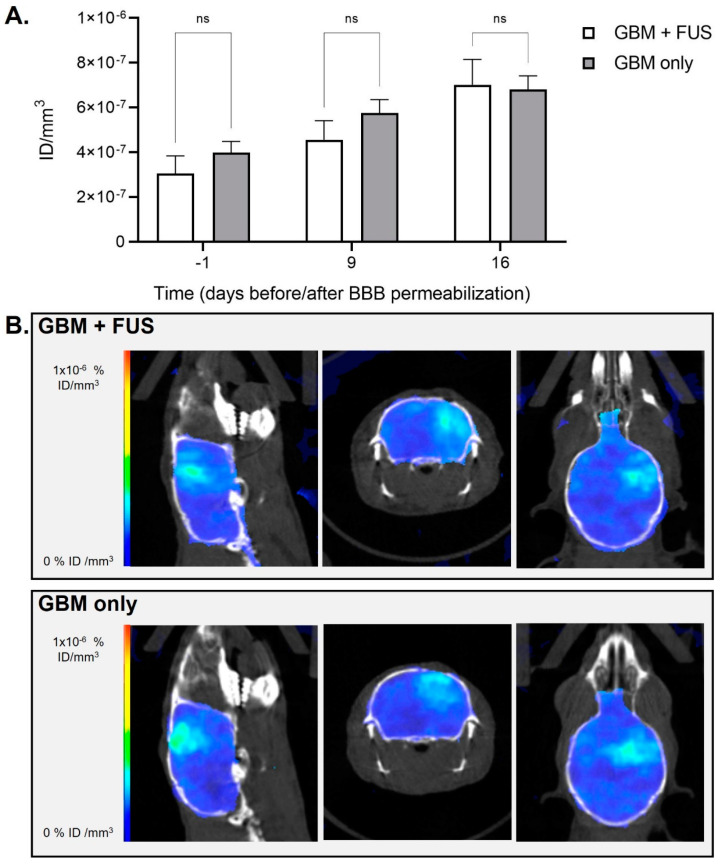
Tumor growth and [68Ga]Ga-RGD_2_ PET/CT imaging following BBB permeabilization by focused ultrasound on tumor-bearing animals. (**A**) Quantification of [68Ga]Ga-RGD_2_ activity in animals of group “GBM + FUS” (tumor-bearing animals undergoing FUS-mediated BBB permeabilization) and group “GBM only” (tumor-bearing animals without FUS-mediated BBB permeabilization) before and after BBB permeabilization. Results are expressed as injected dose per tissue volume (ID/mm^3^; *n* = 5 per group; mean ± SEM; ns: not significant); (**B**) Brain representative images obtained by [68Ga]Ga-RGD_2_-mediated PET/CT imaging at day 16 after hemispheric trajectory focused ultrasound in an animal of group “GBM + FUS” (tumor-bearing animals undergoing FUS-mediated BBB permeabilization) and an animal of group “GBM only” (tumor-bearing animals without FUS-mediated BBB permeabilization).

**Figure 6 pharmaceutics-14-02227-f006:**
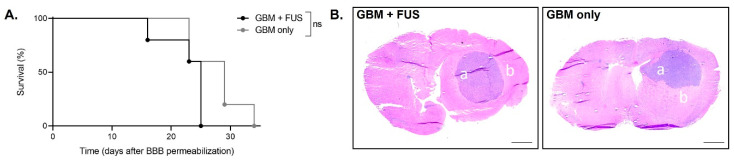
Survival and histological analyses following tumor grafting and BBB permeabilization by focused ultrasound. (**A**) Kaplan-Meier survival curves of the animals of group “GBM + FUS” (tumor-bearing animals undergoing BBB permeabilization) and group “GBM only” (tumor-bearing animals without FUS-mediated BBB permeabilization), *n* = 5 per group; ns: not significant; (**B**) Representative coronal section of Hematoxylin and Eosin staining of tumor-bearing animals that had been exposed or not to FUS-mediated BBB permeabilization (group “GBM + FUS” and “GBM only” respectively) at end point. The a represents relatively low-enhancing glioblastoma tumor cells forming a spheric tumor lesion and the b represents normal brain tissue surrounding the tumor (scale bar: 1 mm).

## Data Availability

Not applicable.

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
