# Peer review of "Molecular Imaging of Ultrasound-Mediated Blood-Brain Barrier Disruption in a Mouse Orthotopic Glioblastoma Model"

_pharmaceutics, 2022, doi:10.3390/pharmaceutics14102227_

Round 1

Reviewer 1 Report

This animal investigation is probably improving the path of knowledge about easing the measurement of effects of US BBB disruption in GBM rodent model, but the way it is described without enough clarity about the pursued goals in the introduction, and the mixture in the text of the "results and discussion of results", makes it difficult to ascertain the real innovation.

I would thank a step forward, explaining the real improvement they achieve in comparison with previous literature and the possible application they envisage to actual clinical practice.

Author Response

We thank the reviewer for this general comment on our manuscript. In the new version we tried to clarify the aims of the study and better stress out our findings. Consequently, the introduction and discussion sections have been significantly amended.

The main objective was to use nuclear imaging to evaluate the effect of BBB opening on GBM-bearing mice. The secondary objectives were the evaluation the impact of BBB-opening on neuroinflammation and tumor growth. Our motivation was to use nuclear imaging as an alternative to magnetic resonance imaging, which is not always available for preclinical studies. Thanks to the variety of tracers in preclinical development for nuclear imaging, this could be used for further characterization of the impact of FUS on inflammation, angiogenesis, tumor and brain microenvironment.

To the best of our knowledge, we are the first to use sequentially and on the same GBM-bearing animal three radiotracers aiming at evaluating BBB permeability ([99mTc]Tc-DTPA), neuroinflammation ([18F]FDG) and tumor growth ([68Ga]Ga-RGD2). This is interesting because DTPA and FDG are currently used in the clinics, which will allow to translate our results to clinical studies aiming at evaluating BBB opening in GBM patients.

However, we are aware that further studies will be needed to increase the gap in knowledge in this field. It could be interesting to confirm our data on more infiltrative and syngeneic animal models such as GL261 in C57BL/6 mice. It could also be interesting to use additional tracers able to target subpopulations of the tumor microenvironment as well as combine our data with immunophenotypic characterization of the brain following BBB-opening in tumor-bearing mice.

Reviewer 2 Report

Summary: Bastiancich and colleagues present a study looking at BBB permeabilisation using focused ultrasound (FUS). They use SPECT/CT and PET/CT imaging to examine FUS effects in mice bearing GBM brain tumours, and in healthy mice. Particularly they focus on the timeline/dynamics of BBB disruption and recovery. Overall I think the study can be suitable for publication. The methods are well done, and the data and analysis appears robust. However, I do see several ways in which the text and figures could be improved. I also think the authors need to describe more about the limitations of their study and to more clearly highlight novelty, impact, significance and purpose.

Detailed comments:

Abstract: I suggest mentioning the type of GBM in mice - i.e. xenograft, mouse GBM etc

Introduction

The introduction is quite good overall. Since Pharmaceutics has a multi-disciplinary readership, including mostly non-specialists in the BBB field, I suggest making the intro a little more “friendly” for non-experts. This means describing some more background information and being clearer about the rationale/purpose of this study.

Around line 45: A couple sentences about how FUS actually works would be very useful

Line 55: I suggest expanding this a little more. The agent does not normally cross the healthy, intact BBB in large concentrations. Hence why it can be used to observe BBB disruption.

I would suggest adding a few sentences about *why* this study is needed, with some relevance to the clinical situation. I.e. what can be learned which could translate to better treatment of human GBM patients.

Materials & Methods: This mostly looks great to me. I appreciate the detail given.

For image quantification, I think a more detail could be given. How were regions/slices selected? Were the people performing measurements blinded to the experimental groups? How many slices were taken per mouse? How many measurements were taken per mouse? Etc etc.

One other weakness: healthy animals are used, not “sham” animals. It would have been better to have a sham which underwent the same drilling, stereotactic injection (of the same EMEM etc) but without tumour cells. I think it’s safe to assume that drilling and injection would induce local disruption the BBB. Thus, some amount of the “GBM” group would be attributable to the surgery itself. This should be mentioned as a limitation in the discussion section.

Figure 2: the use of grey or black statistical marking is not very clear. I suggest simply using other symbols like §§ or ¶¶.

Figure 2A, I am not seeing very obvious difference between the images. Annotating the image to show the spot area would be useful. 

Line 260: I don’t quite understand what is meant by “quantification errors can be performed”

Figure 3: I think the “group1”, “group 2” etc can be more clearly explained in the figure itself. Otherwise one must keep referencing the figure legend. It could simply be “GBM, + FUS”, “GBM only” and “Normal + FUS” or something similar.

In 3A is the tumour region visible in the images? If so, could it be annotated onto the image?

Line 278: The tumour having an impact on BBB permeabilisation shouldn’t be surprising. I think this point could be discussed a little more.

Related to that point, I understand the rationale of showing ipsilateral/contralateral results, but it would also be nice to see the raw numbers, since I expect the GBM mice and healthy mice would have different baselines of BBB permeability. Can these be shown as supplementary data?

Same point for figure 4. The figure shows all 3 groups have the same ipsi/contra ratios. How about the raw signal intensity? I’d expect the GBM mice would uptake more of the reagent.

A weakness of figure 4 is lack of a positive control. If the three experimental groups are negative, it would be best practice to show a positive control when neuroinflammation is induced, thus proving that this technique is measuring neuroinflammation. If the authors do not have these data, I would accept a reference to a study which has proven this technique for measuring neuroinflammation, preferably in brain tumour animals.

Line 296. It may “suggest” lack of neuroinflammation but this method is definitely not conclusive, and I am not convinced that it would be very sensitive. You can suggest other studies in the discussion section which could be done to provider stronger evidence. 

Line 314: The authors refer to “early” stages of tumour development. I see from Fig 6 that some animals start dying after ~15 days and median survival is 20-30 days. Is D7 really “early”? What is the approximate size of the tumour at D7?

Figure 5: Same comment again for group 1/2. It is easier to just say what each group is within the figure.

Figure 6A: Is the difference in survival significant? They can be compared by Mantel-Cox test

Figure 6B: What is this meant to show us? I see a nuclear dense region which is presumably tumour. Is the size/area quantified? (There is also no scale bar on the image). Is haemorrhage area or cell density quantified? How can the authors say “apparent differences in tumor infiltrating behaviour”?

At the very minimum, the authors should show some high magnification crops so that we could see tumour cells, RBCs, glial cells etc. Better, staining such as iba1 could be performed.

Discussion: The “discussion” section (line 332) onwards is quite superficial in my opinion. The authors do mention several other previous related studies. But there is not much comparison of the findings of this study with others. Importantly, there is reflection/discussion of the novelty, impact or limitations/weaknesses of this study. There are several limitations of this study which should be brought to the readers’ attention.

Minor:

Line 238, typo of “protocol” x2

Author Response

We thank the reviewer for her/his positive remarks on our work. In the new version, we tried to better exhibit the aim, novelty, and impact of our experiments.

Round 2

Reviewer 2 Report

All comments have been suitably addressed. The response to reviewers was excellent - clearly written, logical and easy to navigate. Well done.